# Identification of Kynurenic Acid-Induced Apoptotic Biomarkers in Gastric Cancer-Derived AGS Cells through Next-Generation Transcriptome Sequencing Analysis

**DOI:** 10.3390/nu15010193

**Published:** 2022-12-30

**Authors:** Hun Hwan Kim, Sang Eun Ha, Min Yeong Park, Se Hyo Jeong, Pritam Bhagwan Bhosale, Abuyaseer Abusaliya, Chung Kil Won, Jeong Doo Heo, Meejung Ahn, Je Kyung Seong, Hyun Wook Kim, Gon Sup Kim

**Affiliations:** 1Department of Veterinary Medicine, Research Institute of Life Science, Gyeongsang National University, Jinju-daero, Jinju 52828, Republic of Korea; 2Biological Resources Research Group, Gyeongnam Department of Environment Toxicology and Chemistry, Korea Institute of Toxicology, 17 Jegok-gil, Jinju 52834, Republic of Korea; 3Department of Animal Science, College of Life Science, Sangji University, Wonju 26339, Republic of Korea; 4Laboratory of Developmental Biology and Genomics, BK21 PLUS Program for Creative Veterinary Science Research, Research Institute for Veterinary Science, College of Veterinary Medicine, Seoul National University, Seoul 08826, Republic of Korea; 5Division of Animal Bioscience & Intergrated Biotechnology, Jinju 52725, Republic of Korea

**Keywords:** next-generation sequencing, RNA sequencing, AP-1, AGS cells, DEGs

## Abstract

Understanding the triggers and therapeutic targets for gastric cancer, one of the most common cancers worldwide, can provide helpful information for the development of therapeutics. RNA sequencing technology can be utilized to identify complex disease targets and therapeutic applications. In the present study, we aimed to establish the pharmacological target of Kynurenic acid (KYNA) for gastric cancer AGS cells and to identify the biological network. RNA sequencing identified differentially expressed genes (DEGs) between KYNA-treated and untreated cells. A total of 278 genes were differentially expressed, of which 120 genes were up-regulated, and 158 genes were down-regulated. Gene ontology results confirmed that KYNA had effects such as a reduction in genes related to DNA replication and nucleosome organization on AGS cells. Protein–protein interaction was confirmed through STRING analysis, and it was confirmed that cancer cell growth and proliferation were inhibited through KEGG, Reactome, and Wiki pathway analysis, and various signaling pathways related to cancer cell death were induced. It was confirmed that KYNA treatment reduced the gene expression of cancer-causing AP-1 factors (Fos, Jun, ATF, and JDP) in AGS cell lines derived from gastric cancer. Overall, using next-generation transcriptome sequencing data and bioinformatics tools, we confirmed that KYNA had an apoptosis effect by inducing changes in various genes, including factor AP-1, in gastric cancer AGS cells. This study can identify pharmacological targets for gastric cancer treatment and provide a valuable resource for drug development.

## 1. Introduction

Gastric cancer was the fourth leading cause of cancer-related deaths in 2020 and is one of the most common cancers worldwide [1]. It is a malignant tumor that threatens human life and health and is mainly treated through resection [2]. However, chemotherapy is still used in the early stages to relieve symptoms and suppress metastasis [3]. Cisplatin and fluorouracil, used primarily as chemotherapeutic agents, cause various side effects in patients, including vomiting and hemotoxicity [4]. Therefore, there is a need to develop a therapeutically effective drug with low toxicity to treat gastric cancer.

Among approaches for cancer treatment, it is essential to understand changes in the molecular pathways and signaling mechanisms of tumor cells through drugs [5]. When normal cells become cancerous, various genes and proteins are involved in the transformation [6]. Although it is relatively difficult to identify the mechanism of action according to drug treatment, next-generation drugs focus on developing methods targeting specific genes and proteins [7]. Obtaining information about the target and its mechanism of action can provide precise insight into the treatment of the disease. In addition, targeted treatment methods can minimize cytotoxicity and side effects compared to conventional anticancer drugs [8]. It has been confirmed that tumors are caused by various factors, including genes, proteins, and metabolites [9].

Network pharmacology is a study that utilizes computer network analysis and biomedical data, and helps to discover the underlying mechanisms between drugs and their target agents [10]. Transcriptome analysis is a method to identify differentially expressed genes during drug treatment using bioinformatics, and is a technique that helps to understand the discovery of essential genes and the mechanism of action of drugs [11]. It is possible to quickly identify differentially expressed genes and secure a target to accelerate treatment. However, it is still a problematic study to predict the onset of gastric cancer and to identify target factors for early treatment and treatment. Therefore, there is a need for a process to acquire more target factors to improve treatment.

The AP-1 complex is a dimeric transcription factor including Jun (c-Jun, JunB, and JunD), Fos (c-Fos, FosB, Fra2, and FosL), and ATF (ATF2, ATF3, and ATF7), and regulates a variety of cellular processes, including apoptosis, survival, migration, and differentiation [12,13]. It has been reported that C-Fos, together with interleukin-1 receptor type 2 (IL1R2), promotes angiogenesis in human colon cancer cells to promote tumor progression [14]. Therefore, AP-1 is an essential target for cancer treatment, and the related mechanism is being studied [15]. Therefore, this suggests that drug-mediated alteration of the corresponding biomarkers can control a variety of cells, including cancer migration, growth, and proliferation.

Kynurenic acid (KYNA) is a phenolic compound with a phenol ring and various physiological activities. It is a metabolite of the kynurenine pathway, which is the catabolic process of tryptophan [16]. KYNA has been reported as an inhibitor of p38 in renal cancer cells and renal cell carcinoma Caki-2 cells [17], and it inhibits the proliferation of cancer cells by affecting other signaling factors such as MAPK, ERK, and AKT turnout [16]. The present study performed transcriptome analysis and network pharmacology analysis to identify potential targets of KYNA for gastric cancer cell AGS. We discovered AP-1 as an anti-tumor regulator that KYNA induces significant changes in AGS cells and secured primary data on changes in other genes.

## 2. Materials and Methods

### 2.1. Cell Culture

AGS human gastric cancer cells (Korea Cell Line Bank, Seoul, Republic of Korea) were cultured in RPMI medium (Gibco; BRL Life Technologies, Grand Island, NY, USA) containing 10% fetal bovine serum (FBS) (Gibco; BRL Life Technologies, Grand Island, NY, USA) and 1% penicillin/streptomycin (P/S) (Gibco; BRL Life Technologies, Grand Island, NY, USA) at 37 °C in a humidified atmosphere of 5% CO_2_. Kynurenic acid (KYNA) was purchased from Sigma-Aldrich Corp. (St. Louis, MO, USA).

### 2.2. Isolation of RNA for Sequencing

AGS cells were seeded into 60 mm plates at 1 × 10^6^ and treated with 250 μM of Kynurenic acid (KYNA) for 24 h at 37 °C. After incubation, total RNAs were extracted using TRIzol. A spectrophotometer was used to measure the amount of RNA. The sequencing of isolated total RNA was then used to determine the expression levels.

### 2.3. Library Preparation and Sequencing

According to the following methodology, the mRNA sequencing was prepared by Theragenbio (Seongnam-si, Gyeonggi-do, Republic of Korea). We created 151 bp paired-end sequencing libraries using the TruSeq stranded mRNA Sample Preparation Kit (Illumina, CA, USA). Utilizing oligo (dT) magnetic beads, mRNA molecules were specifically isolated and fragmented from 1 μg of total RNA. Through random hexamer priming, single-stranded cDNAs were created from the fragmented mRNAs. Double-stranded cDNA was created by using this as a template for second-strand synthesis. After end repair, A-tailing and adapter ligation were completed in order, and cDNA libraries were amplified using PCR (Polymerase Chain Reaction). The Agilent 2100 BioAnalyzer (Agilent, CA, USA) was used to assess these cDNA libraries’ quality. According to the manufacturer’s library quantification methodology, they were measured using the KAPA library quantification kit (Kapa Biosystems, MA, USA). After cluster amplification of denatured templates, Illumina NovaSeq6000 (Illumina, CA, USA) was used to advance the sequencing process as paired-end (2 × 151 bp).

### 2.4. Transcriptome Data Analysis

#### 2.4.1. Filtering and Sequence Alignment

Adapter sequences and read ends with a Phred quality score of less than 20 were deleted throughout the filtering process, and reads less than 50 bp were also eliminated using cutadapt v.2.8. Following ENCODE standard choices and the “quantMode TranscriptomeSAM” option (refer to “Alignment” of “Help” section in the Html report) for estimation of transcriptome expression level, filtered reads were mapped to the reference genome associated with the species using the aligner STAR [18].

#### 2.4.2. Gene-Expression Estimation

By utilizing the option “strandedness”, the RSEM v.1.3.1 [19] program estimated gene expression while taking into account the orientation of the reads about the library protocol. The “estimate-rspd” option was used to increase measurement accuracy. The default settings were used for all other parameters. Values for FPKM and TPM were generated to standardize depth across samples.

#### 2.4.3. Gene Ontology (GO) Analysis

The GO database was used to analyze biological process (BP), cellular component (CC), and molecular function (MF). A GO-based trend test was performed for DEG functional characterization using the Wallenius non-central hypergeometric distribution and the R package GOseq [20]. Following the test, selected genes with a *p*-value of 0.05 or below were considered statistically significant.

#### 2.4.4. Differentially Expressed Gene (DEG) Analysis

Using the TCC v.1.26.0 [21] R package, DEGs were found based on the projected read counts from the previous phase. The TCC package compares tag-count data using effective normalizing techniques. Normalization factors were computed using the iterative DESeq2 [22]/edgeR [23] technique. Using the p.adjust function of the R package with the default parameter values, the Q-value was determined based on the *p*-value. A *q*-value criterion of less than 0.05 was used to identify the DEGs and adjust for multiple testing mistakes.

### 2.5. STRING Network and Enrichment Pathway Analysis

The protein–protein interaction (PPI) for the top 30 genes among up-regulated and down-regulated DEGs was performed using an online tool Search Tool for the Retrieval of Interacting Genes/Proteins (STRING, version 9.1) (http://string-db.org (accessed on 8 February 2022)). The PPI score was set based on medium confidence (0.4). In the PPI network, each node depicts a protein, while the edges represent the strength of the relationship between proteins. Likewise, for the top 30 DEGs, all functionally enriched pathways of significantly expressed genes were determined using the Kyoto Encyclopedia of Genes and Genomes (KEGG), Reactome, and Wiki pathway databases. Additionally, annotated keywords were also identified.

### 2.6. Drug and Disease Association Analysis

Drug and disease association analyses were performed using WEB-based GEne SeT AnaLysis (WebGestalt), which is an online toolkit for functional genomic enrichment. The significance was set at FDR < 0.05, and the gene count was set at ≥5.

### 2.7. Molecular Docking Analysis

To perform molecular docking analysis, we retrieved the protein structure from PDB (https://www.rcsb.org/, accessed on 1 May 2021) using the search ID 5VPD (AP-1), and the compound structures (Dacarvazine and Kynurenic acid) were downloaded from PubChem (https://pubchem.ncbi.nlm.nih.gov/, accessed on 1 May 2021). Docking analysis was performed using pyMol and USCF chimera with the default parameters. The affinity of the binding is determined using estimated free energy binding and total intermolecular energy.

### 2.8. Analysis of Protein Expression by Western Blot

AGS cells were seeded into 60 mm plates at 4 × 10^5^ cells per well and treated with the indicated concentrations of KYNA of (0, 150, 200, and 250) μM for 24 h. Cells were lysed using radioimmunoprecipitation assay (RIPA) buffer (iNtRON Biotechnology, Seoul, Republic of Korea) containing phosphatase and protease inhibitor cocktail (Thermo Scientific, Rockford, IL, USA). Protein concentrations were determined using a Pierce™ BCA assay (Thermo Fisher Scientific, Rockford, IL, USA). An equal quantity of protein (10 μg) from each sample was electrophoresed on (8–15%) SDS-polyacrylamide gels and transferred to a polyvinylidene difluoride (PVDF) membrane (ATTO Co., Ltd., Tokyo, Japan), and then the membrane was incubated with the primary antibodies followed by a conjugated secondary antibody to peroxidase. The obtained proteins were detected by an electrochemiluminescence (ECL) detection system (Bio-Rad Laboratory, Hercules, CA, USA), and analyzed using the Image Lab 4.1 (Bio-Rad) program. The densitometry readings of the protein bands were normalized by comparison with the expression of β-actin as control, using the ImageJ software program (U.S. National Institutes of Health, Bethesda, MD, USA). Antibodies of c-Fos (Cat. #2250S), c-Jun (Cat. #9165S), p-c-Fos (Cat. #5348S), and p-c-Jun (Cat. #9261S) were purchased from Cell Signaling Technology (Danvers, Ma, USA).

### 2.9. Statistical Analysis

All the experimental data were analyzed using GraphPad Prism version 8.0.2 (GraphPad Software). The results were expressed as the means ± standard deviation (SD). Results were evaluated using the Student’s *t*-test, and a *p*-value < 0.05 was considered statistically significant.

## 3. Results

### 3.1. Identification of Genes and DEGs

It was confirmed in a previous study [24] that KYNA treatment induces tumor cell death in AGS cells. A total of 278 DEGs (log_2_ (Fold Change) > 1.0 and *p*-value < 0.05) were identified in the KYNA treatment group, of which 120 up-regulated genes and 158 down-regulated genes were identified (Table 1). Additionally, the R-Bioconductor volcanic pilot revealed DEGs among the total genes (Figure 1).

### 3.2. Functional and Enrichment Analysis

Gene ontology was used to identify a representative class of genes or proteins that were excessively changed by KYNA in AGS cells. Gene groups corresponding to molecular functions, biological processes, and cellular components were analyzed (Figure 2 and Table 2). As a result of molecular function, the structural constituent of the extracellular matrix was the most decreased, followed by a decrease in protein heterodimerization activity, showing a tendency to decrease in genes related to cell differentiation and growth. The biological process also showed a decrease in genes involved in DNA replication and nucleosome construction in cells with a similar trend. In the cellular component, genes related to extracellular regions and neuronal signaling were decreased. This suggests that KYNA suppressed the expression of genes involved in the growth and proliferation of AGS cells.

The top 10 genes with high significance are listed with their function and z-score of the respective term.

### 3.3. Protein–Protein Interaction (PPI) and Enrichment Pathway of DEGs

Protein–protein interaction (PPI) network analysis was constructed using the STRING database for the top 30 differentially expressed genes (DEGs) up and down (Figure 3), and the related enrichment pathways were analyzed (Table 3). The results of the PPI STRING analysis of up-regulated genes were divided into two groups, and down-regulated genes were divided into three groups. Up-regulated DEGs were involved in various apoptoses such as in the PI3K-AKT signaling pathway and the Ras signaling pathway in the KEGG pathway analysis, and DNA damage-related results were also obtained in the Wiki pathway. In the Reactome pathway, pathways related to cell signal transduction were identified, and in the annotated keywords, results related to DNA damage, repair, and recombination were also confirmed. This suggests that KYNA induces apoptosis of AGS cells through the up-regulation of up-regulated genes. KEGG, wiki, and Reactome pathways in down-regulated genes confirmed the results of metabolic processes and DNA replication related to cell proliferation. This suggests that KYNA down-regulated genes are involved in AGS cell metabolism and DNA replication.

### 3.4. Expression Comparison of AP-1 Factors and Molecular Docking with KYNA

The expression of AP-1 factors related to tumor growth and proliferation was confirmed (Table 4). Fos, Jun, JDP (Jun dimerization protein), and ATF (Activating transcription factor) constituting AP-1 were decreased upon KYNA treatment, suggesting that KYNA inhibits the growth and proliferation of AGS cells. Additionally, ligand–protein docking analysis was performed through UCSF Chimera software. Both ligands AP-1 and KYNA were found to occupy the active site, as illustrated in Figure 4. In addition, many active sites have been shown to aid ligand binding. Active sites involved in the binding of AP-1 to KYNA were identified as ALA281, ALA168, ALA169, ALA185, GLU192, ASN165, LYS166, GLN189, GLU191, CYS285, LYS284, ARG288, ARG164, and ASP188 (Table 5). The molecular binding energy score was found to be −6.3 kcal/mol.

The table shows the list of interacting amino acids and their binding energy.

### 3.5. Validation of AP-1 Factor Expression Using Western Blot Analysis

Western blot analysis of AGS cells during KYNA treatment confirmed the expression of c-Fos and c-Jun, which are AP-1 factors (Figure 5). It was confirmed that the expression of p-c-Fos and p-c-Jun proteins decreased by inhibiting phosphorylation of c-Fos and c-Jun proteins during KYNA treatment. This result was consistent with the decreased expression of AP-1 factors obtained through transcriptome analysis, further proving the role of KYNA in regulating the expression of important factors in AGS cells.

### 3.6. Therapeutic Drug and Disease Association and Molecular Docking Analysis

Through drug and disease association analysis, the top 10 drugs significantly related to differentially regulated genes were selected (Table 6). Among the selected drugs, Dacarbazine was approved by the FDA (Abbreviated New Drug Application (ANDA) number; 075371, Fresenius, Kabi, Illinois, IL, USA), and it was confirmed that it was used as an anticancer drug. Drug and protein molecular docking were performed to confirm the molecular binding of Dacarbazine to AP-1 (Figure 6). In ligand–protein docking using UCSF Chimera software, it was found that a binding site was shared between the two factors, as illustrated in the figure. In addition, various active sites have been shown to aid in ligand uptake. Mutual binding amino residues were found to be GLU191, GLU193, GLU192, ASP188, GLN189, ALA185, ARG288, LYS166, ASN165, ALA169, LYS284, CYS285, CYS172, ALA168, and ALA281 (Table 7), with a molecular binding score of −5.2 kcal/mol demonstrating the molecular binding energy of autodock.

The table shows the list of interacting amino acids and their binding energy.

## 4. Discussion

Understanding disease-causing factors provides a roadmap for the development of new therapeutic drugs [25]. RNA sequencing technology is used for disease diagnosis and therapeutic applications by identifying the targets of various diseases, including cancer [26]. Complex relationships can be understood through network analysis, usually using bioinformatics and pharmacology to help elucidate mechanisms between drugs and their targets [27]. Through this, targeted drug treatment by identifying tumor-specific expressed genes can provide patients with a variety of treatment options.

The effect of KYNA on the genetic change in AGS cells was analyzed using RNA seq. A total of 60,676 genes were identified, and gene ontology changes (Figure 2) and differentially expressed genes (DEGs) were confirmed between the KYNA-treated group and the KYNA-untreated group (Figure 1). Genes related to cell differentiation and growth decreased in molecular function, genes related to DNA replication and nucleosome construction decreased in biological process, and genes related to synaptic signal transduction decreased in cellular components (Table 2). Through GO analysis, it was confirmed that KYNA had a negative effect on tumor cells.

The association of 30 differentially expressed genes (DEGs) was confirmed through protein–protein interaction (PPI) network analysis. The up-regulated genes showed interaction in a total of two groups, and the down-regulated genes consisted of a total of three groups (Figure 3). Up-regulated interacting genes affected various signaling pathways, including the PI3K-AKT signaling pathway in the KEGG pathway. In the Wiki pathway, signaling pathways related to the ATM signaling pathway and DNA damage were also shown, and in the Reactome pathway, results such as cell-cycle checkpoint and DNA repair were shown. Down-regulated genes obtained results related to cell proliferation, metabolism, and DNA replication in the KEGG, wiki, and Reactome pathways (Table 3). These results suggest that KYNA induces apoptosis of AGS.

AP-1 [28], which affects various regulatory processes including tumor growth, proliferation, migration, cell cycle, and apoptosis, has members Jun, Fos, and ATF [29]. In particular, Fos and Jun are involved in DNA synthesis and G0-G1 during the cell cycle and are known to generate CD8 T-lymphocytes of T-cells [30]. In addition, c-Fos expression in gastric cancer cells is associated with lymph node metastasis, invasion, and short survival, and has been reported to have a poor prognosis [31]. In this study, as a result of analyzing the expression level of AP-1 members, it was confirmed that Fos, Jun, ATF, and JDP were decreased upon KYNA treatment (Table 4). Additionally, as a result of confirming the binding affinity of KYNA and the AP-1 complex (Jun and Fos) through molecular docking analysis, a binding force of −6.3 kcal/mol was confirmed (Figure 4 and Table 5). In addition, as a result of analyzing the protein expression of Jun and Fos through Western blot analysis, it was confirmed that AP-1 factors decreased in a concentration-dependent manner when KYNA was treated (Figure 5). This suggests that KYNA treatment in AGS cells suppresses the expression of AP-1 factors, thereby inhibiting tumor growth and proliferation. The top 10 drugs related to DEGs were obtained through drug and disease association analysis (Table 6). Among them, Dacarbazine was approved by the FDA and confirmed to be used as an anticancer agent [32]. As a result of confirming the binding affinity between Dacarbazine and the AP-1 complex (Jun and Fos) through molecular docking analysis, a binding force of −5.2 kcal/mol was shown (Figure 6 and Table 7). AP-1, which induces tumor growth and proliferation, showed a somewhat higher binding force with KYNA than Dacarbazine, which is currently used as a drug. This suggests that KYNA induces inhibition of AP-1.

KYNA is a physiologically active ingredient found in various herbs [33] which induces apoptosis through cell-cycle regulation and signaling pathways in colorectal and renal cancer cells [17]. It has also been reported to inhibit cell proliferation and growth through the PI3K/AKT and MAPK signaling pathways in adenocarcinoma [16]. It has been demonstrated that KYNA has a direct effect on the gastric wall to prevent gastric ulcers [34]. In a previous study, KYNA was confirmed to induce apoptosis in gastric cancer-derived AGS cells [24]. In this study, it was confirmed that KYNA inhibits the growth and proliferation of cancer cells through the identification of biomarkers. These results suggest that KYNA can have an anti-tumor effect in gastric cancer, and it is judged that it can be used as basic data for the development of gastric cancer therapeutics.

## 5. Conclusions

Transcriptome analysis results provide potential insights into tumor therapeutics in KYNA-treated AGS cells. In this study, it was found that KYNA induces the expression of genes that responds to apoptosis and, in particular, up-regulates the expression of AP-1 factors. In addition, the results of this study show the effect of KYNA on the gene expression of AGS cells, and provide a novel research method that can be used to confirm the effects of other drugs on other cancer cells. Results suggest that KYNA can be considered for the development of treatment for gastric cancer.

## Figures and Tables

**Figure 1 nutrients-15-00193-f001:**
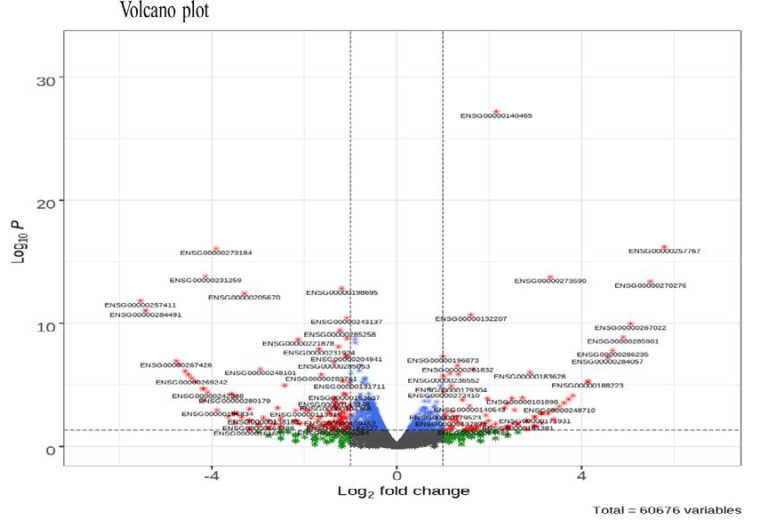
**Volcano plot of differentially expressed genes (DEGs).** A total of 60,676 variables were considered for the plot. The fold-change was plotted based on log_2_ FC and *p*-value.

**Figure 2 nutrients-15-00193-f002:**
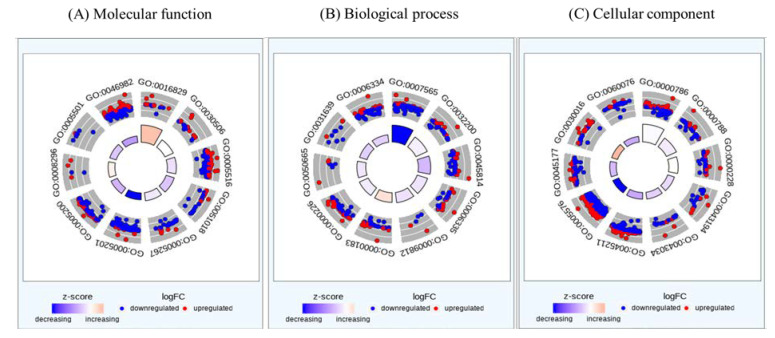
**Circos plot representation of gene enrichment ontology analysis plotted in terms of molecular function, biological process, and cellular component from differentially regulated genes of KYNA against AGS cells.** Blue: down-regulated, Red: up-regulated.

**Figure 3 nutrients-15-00193-f003:**
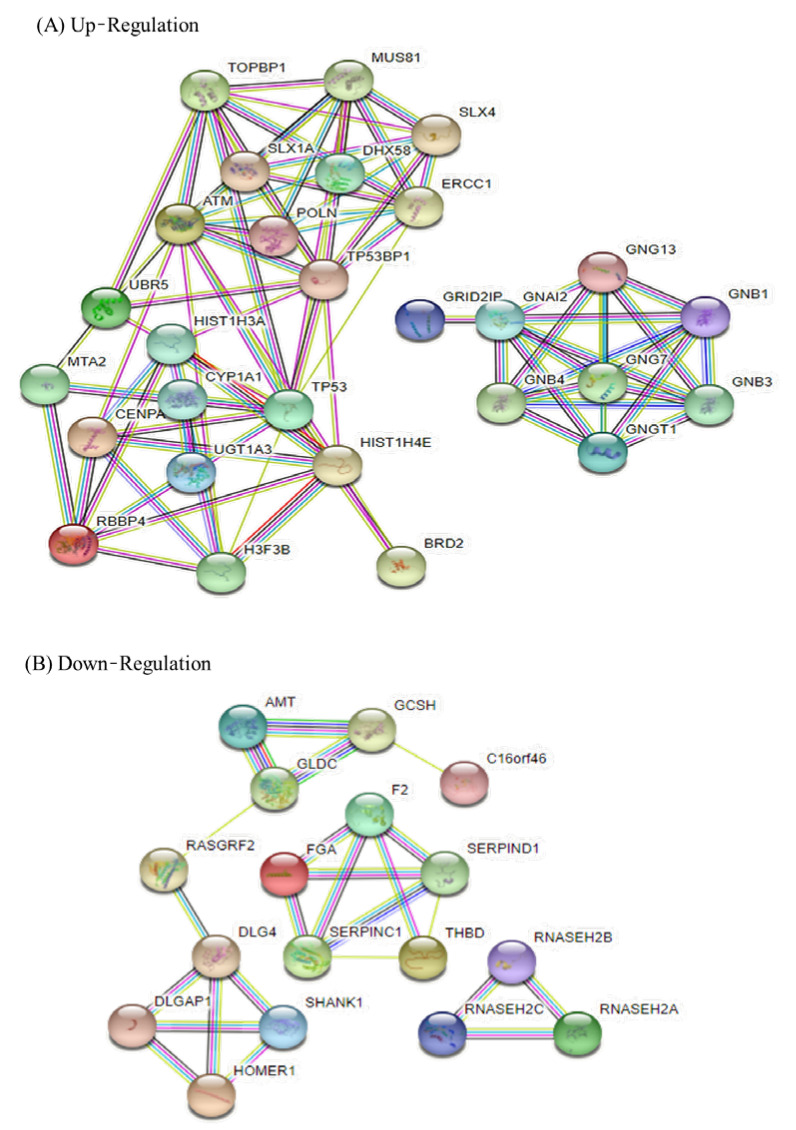
**Protein—protein interaction (PPI) analysis of up-regulation and down-regulation DEGs analyzed by STRING software.** The network nodes represent genes (showing the interactions), and the round nodes denote individual genes. The line color indicates the type of interaction evidence.

**Figure 4 nutrients-15-00193-f004:**
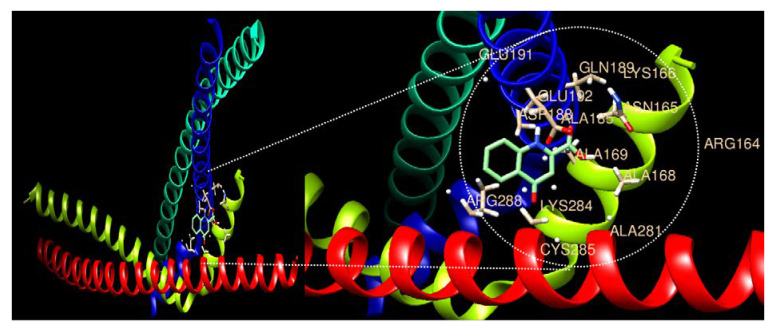
**Molecular docking analysis of the AP-1 complex (c-Jun and c-Fos) and Kynurenic acid.** The 3D structure of AP-1 factors is bound efficiently with Kynurenic acid.

**Figure 5 nutrients-15-00193-f005:**
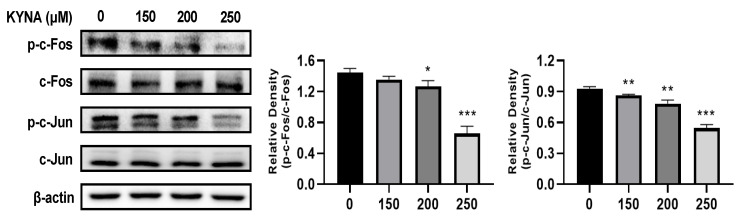
**Western blot validation of crucial targets of KYNA against AGS cells.** Protein expression of crucial proteins c-Fos and c-Jun on KYNA-treated AGS cells for 24 hr. The results obtained from three independent experiments are expressed as mean ± standard deviation (SD) compared with the control group. * *p* < 0.05, ** *p* < 0.01, *** *p* < 0.001.

**Figure 6 nutrients-15-00193-f006:**
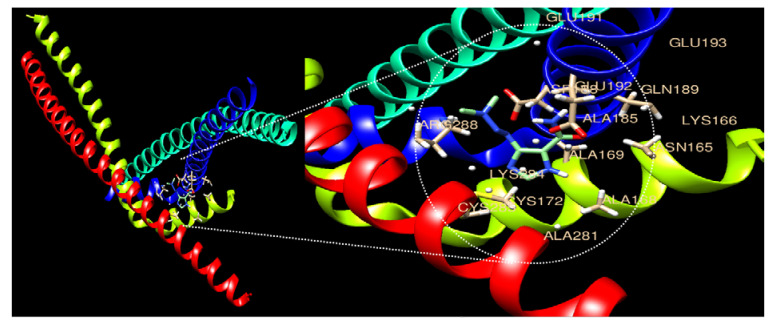
**Molecular docking analysis of the AP-1 complex (c-Jun and c-Fos) and Dacarbazine.** The 3D structure of AP-1 factors is bound efficiently with the drug Dacarbazine.

**Table 1 nutrients-15-00193-t001:** Sequencing statistics data.

No	Name	Type	Reads	Bases	Bases (Gb)	GC	N	Q30
1	Control	Raw	41,131,670 (100%)	6,210,882,170 (100%)	6.21	3,103,637,588 (49.97%)	33,918 (0%)	5,558,401,799 (89.49%)
1	Control	Clean	39,692,472 (96.5%)	5,845,982,692 (94.12%)	5.85	2,891,132,459 (49.46%)	32,209 (0%)	5,314,657,230 (90.91%)
2	Test	Raw	43,673,272 (100%)	6,594,664,072 (100%)	6.59	3,299,521,366 (50.03%)	35,186 (0%)	6,041,568,334 (91.61%)
2	Test	Clean	42,815,932 (98.04%)	6,224,436,097 (94.39%)	6.22	3,100,915,971 (49.82%)	34,024 (0%)	5,764,074,214 (82.6%)

Reads: Number of reads (Reads/Raw reads × 100); Bases: Number of bases (Bases/Raw bases × 100); Bases (Gb): Number of bases (in Giga base unit); GC: Number of G and C bases (GC/Bases × 100); N: Number of N bases (N/Bases × 100); Q30: Number of over Q30 bases (Q30: 99.9% Base Call Accuracy) (Q30/Bases × 100); Q20: Number of over-Q20 bases (Q20: 99% Base Call Accuracy) (Q20/Bases × 100).

**Table 2 nutrients-15-00193-t002:** List of enriched gene ontology in terms of molecular function, biological process, and cellular component.

**(A) Molecular Function**		
**Gene ID**	**Term**	**z-score**
GO:0005201	extracellular matrix structural constituent	−5.422
GO:0046982	protein heterodimerization activity	−2.366
GO:0005200	structural constituent of cytoskeleton	−1.732
GO:0051018	protein kinase A binding	−1.342
GO:0005501	retinoid binding	−1.342
GO:0005516	calmodulin binding	−0.728
GO:0005267	potassium channel activity	−0.535
GO:0030506	ankyrin binding	0
GO:0008296	3′-5′-exodeoxyribonuclease activity	0.447
GO:0016829	lyase activity	1.732
**(B) Biological Process**		
**Gene ID**	**Term**	**z-score**
GO:0007565	female pregnancy	−3.457
GO:0045814	negative regulation of gene expression, epigenetic	−1.043
GO:0031639	plasminogen activation	−0.707
GO:0006334	nucleosome assembly	−0.688
GO:0009812	flavonoid metabolic process	−0.447
GO:0006335	DNA replication-dependent nucleosome assembly	−0.408
GO:0050665	hydrogen peroxide biosynthetic process	−0.378
GO:0000226	microtubule cytoskeleton organization	−0.305
GO:0032200	telomere organization	−0.229
GO:0000183	chromatin silencing at rDNA	0.557
**(C) Cellular Component**		
**Gene ID**	**Term**	**z-score**
GO:0005576	extracellular region	−4.899
GO:0045211	postsynaptic membrane	−1.841
GO:0060076	excitatory synapse	−1.807
GO:0045177	apical part of cell	−1.457
GO:0043034	costamere	−1.0
GO:0043194	axon initial segment	−0.775
GO:0000788	nuclear nucleosome	−0.408
GO:0000786	nucleosome	−0.119
GO:0000228	nuclear chromosome	0.0
GO:0030016	myofibril	1.789

**Table 3 nutrients-15-00193-t003:** KEGG, Wiki, and Reactome pathways and annotated keyword classification and functional enrichment for the up-regulation and down-regulation DEGs.

**Up-regulation DEGs**			
**KEGG Pathways**	**Description**	**Strength**	**FDR**
hsa04014	Ras signaling pathway	1.27	0.0000138
hsa04151	PI3K-AKT signaling pathway	1.15	0.00000999
hsa04218	Cellular senescence	1.15	0.0194
hsa05202	Transcriptional misregulation in cancer	1.09	0.0268
hsa04727	GABAergic synapse	1.75	0.00000000998
hsa04926	Relaxin signaling pathway	1.58	0.0000000247
hsa04728	Dopaminergic synapse	1.58	0.0000000247
hsa04371	Apelin signaling pathway	1.57	0.0000000247
hsa04062	Chemokine signaling pathway	1.42	0.000000192
**Wiki pathways**	**Description**	**Strength**	**FDR**
WP1545	miRNAs involved in DNA damage response	1.97	0.0149
WP2516	ATM signaling pathway	1.73	0.002
WP4016	DNA IR-damage and cellular response via ATR	1.66	0.0000131
WP3959	DNA IR-double-strand breaks and cellular response via ATM	1.59	0.0045
WP4172	PI3K-AKT signaling pathway	1.16	0.0000347
WP3932	Focal adhesion: PI3K-AKT-mTOR signaling pathway	1.14	0.0003
**Reactome pathways**	**Description**	**Strength**	**FDR**
HAS-392851	Prostacyclin signaling through prostacyclin receptor	2.34	0.000000000201
HAS-8964616	G beta: gamma signaling through CDC42	2.32	0.000000000227
HAS-418217	G beta: gamma signaling through PLC beta	2.32	0.000000000227
HAS-202040	G-protein activation	2.24	0.0000000000383
HAS-392451	G beta: gamma signaling through PI3Kgamma	2.22	0.000000000441
HAS-6803204	TP53 regulates transcription of genes involved in cytochrome C release	1.87	0.0096
HSA-69473	G2/M DNA damage checkpoint	1.55	0.00018
HSA-2559580	Oxidative stress-induced senescence	1.35	0.0082
HSA-73894	DNA repair	1.31	0.0000000205
HSA-69620	Cell-cycle checkpoints	1.11	0.0012
**Annotated keywords**	**Description**	**Strength**	**FDR**
KW-0233	DNA recombination	1.45	0.0316
KW-0255	Endonuclease	1.43	0.0316
KW-0013	ADP-ribosylation	1.32	0.0406
KW-0234	DNA repair	1.24	0.00000466
KW-0227	DNA damage	1.22	0.00000156
**Down-regulation DEGs**			
**KEGG pathways**	**Description**	**Strength**	**FDR**
hsa00630	Glyoxylate and dicarboxylate metabolism	2.06	0.00043
hsa03030	DNA replication	1.98	0.00043
hsa00260	Glycine, serine, and threonine metabolism	1.96	0.00043
hsa04610	Complement and coagulation cascades	1.85	0.00000307
hsa4724	Glutamatergic synapse	1.62	0.00043
hsa01200	Carbon metabolism	1.47	0.008
**Wiki pathways**	**Description**	**Strength**	**FDR**
WP4705	Pathways of nucleic acid metabolism and innate immune sensing	2.33	0.00018
WP4875	Disruption of postsynaptic signaling by CNV	2.03	0.00079
**Reactome pathways**	**Description**	**Strength**	**FDR**
HAS-140837	Intrinsic pathway of fibrin clot formation	2.06	0.02940
HAS-442982	Ras activation upon Ca^2+^ influx through NMDA receptor	1.57	0.0011
HAS-8957275	Post-translational protein phosphorylation	1.51	0.024
**Annotated keywords**	**Description**	**Strength**	**FDR**
KW-0225	Disease mutation	0.56	0.0102

FDR: False-discovery rate.

**Table 4 nutrients-15-00193-t004:** Expression comparison of AP-1 factors.

Gene ID	Symbol	RPKM	
		C	T
ENSG00000075426	FOS L2	14.98	13.7
ENSG00000125740	FOS B	0.6	0.34
ENSG00000130522	JUN D	53.29	48.77
ENSG00000170345	FOS	5.06	4.84
ENSG00000171223	JUN B	21.87	19.96
ENSG00000175592	FOS L1	24.33	22.91
ENSG00000177606	JUN	51.06	48.78
ENSG00000140044	JDP 2	2.41	2.17
ENSG00000115966	ATF 2	11.32	10.56
ENSG00000170653	ATF 7	9.38	8.82

RPKM: reads per kilobase of per million mapped reads. C: without KYNA, T: with KYNA 250 µM.

**Table 5 nutrients-15-00193-t005:** Molecular docking studies of Kynurenic acid with the AP-1 complex and their binding energy.

Compound-Protein	Interacting Amino Acid Residues	Final Intermolecular Energy
Kynurenic acid	ALA281, ALA168, ALA169, ALA185, GLU192, ASN165, LYS166, GLN189, GLU191, CYS285, LYS284, ARG288, ARG164, ASP188	−6.3 kcal/mol

**Table 6 nutrients-15-00193-t006:** Results of drug and disease association analysis.

Gene Set	Description	Size	Expect	Ratio	*p*-Value
DB04953	Ezogabine	10	0.017071	117.16	0.00010437
DB01095	Fluvastatin	18	0.030727	65.089	0.0035292
DB00586	Diclofenac	28	0.047798	41.843	0.00086596
DB00197	Troglitazone	30	0.051212	39.053	0.00099517
DB00633	Dexmedetomidine	5	0.0085353	117.16	0.0085120
DB00740	Riluzole	5	0.0085353	117.16	0.0085120
**DB00851**	**Dacarbazine**	**5**	**0.0085353**	**117.16**	**0.0085120**
DB00889	Granisetron	5	0.0085353	117.16	0.0085120
DB04905	Tesmilifene	5	0.0085353	117.16	0.0085120
DB06089	ICA-105665	5	0.0085353	117.16	0.0085120

A list of enriched drugs concerning differentially expressed gene sets is depicted. The top 10 drugs were chosen based on their significance rate.

**Table 7 nutrients-15-00193-t007:** Molecular docking studies of Dacarbazine with the AP-1 complex and their binding energy.

Compound-Protein	Interacting Amino Acid Residues	Final Intermolecular Energy
Dacarbazine	GLU191, GLU193, GLU192, ASP188, GLN189, ALA185, ARG288, LYS166, ASN165, ALA169, LYS284, CYS285, CYS172, ALA168, ALA281	−5.2 kcal/mol

## Data Availability

The data used to support the findings of this study are available upon request from the corresponding author.

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
