# Peer review of "Identification of Kynurenic Acid-Induced Apoptotic Biomarkers in Gastric Cancer-Derived AGS Cells through Next-Generation Transcriptome Sequencing Analysis"

_nutrients, 2022, doi:10.3390/nu15010193_

Round 1

Reviewer 1 Report (New Reviewer)

The present article identified kynurenic acid-induced apoptotic biomarkers in gastric cancer-derived AGS cells through transcriptome sequencing analysis. The finding is important for understanding the antitumor mechanism of kynurenic acid against gastric cancer. I would like to suggest to accept it after minor revision on English writing, specifically the grammar. Meanwhile, please state out the novelty in the conclusion.

For example, in the conclusion,

1.     P14, line 353, “it was found that KYNA induces the expression of genes that induce apoptosis,” two “induce”s were used in same sentence.  “was found” vs “induces “? Suggest to be revised as “it was found that KYNA induced the expression of genes that responds to apoptosis,”

2.     P14, line 354, “increases the expression of AP-1 factors”, suggest to be revised as “up-regulates the expression of AP-1 factors”

Author Response

For reviewer

Thank you for your attentive comments.  

  • Meanwhile, please state out the novelty in the conclusion.

Response : Thank you for comment. We added that content.

  • P14, line 353, “it was found that KYNA induces the expression of genes that induce apoptosis,” two “induce”s were used in same sentence. “was found” vs “induces “? Suggest to be revised as “it was found that KYNA induced the expression of genes that responds to apoptosis,”

Response : Thank you for comment. We have corrected it.

  • P14, line 354, “increases the expression of AP-1 factors”, suggest to be revised as “up-regulates the expression of AP-1 factors”

Response : Thank you for comment. We added that content.

Additional English corrections were made.

Thanks you for your detail comment.

Reviewer 2 Report (New Reviewer)

this is a very nice work

I am only concern that the meaning of Identification of Kynurenic Acid-Induced Apoptotic Biomarkers in Gastric Cancer-Derived AGS cells.

It should be Identification of Kynurenic Acid-Induced Apoptotic RELATED Biomarkers in Gastric Cancer-Derived AGS cells.

Author should explain in introduction relation of these biomarkers to apoptosis.

Author Response

For reviewer

Thank you for your attentive comments.  

  • Author should explain in introduction relation of these biomarkers to apoptosis.

Response : Thank you for your comment. We added that content.

Thanks you for your detail comment.

This manuscript is a resubmission of an earlier submission. The following is a list of the peer review reports and author responses from that submission.

Round 1

Reviewer 1 Report

This study aims to analyse, by Next- Generation Transcriptome Sequencing Analysis, potential genes involved in apoptosis induction by Kynurenic acid in in gastric cancer-derived AGS. The design does appear well realized. There are other well-known apoptosis inducing agents as control and a limited dose effect curve is showed as well. By this approach, Authors showed differentially expressed genes (DEGs) between Kynurenic acid -treated and untreated cells. Specifically, a total of 278 genes were differentially expressed, of which 120 genes were up-regulated, and 158 genes resulted down-regulated.

At present, RNA sequencing technology is used for disease diagnosis and to highlight potential therapeutic applications by identifying the targets of various diseases, including cancer. in fact, as stressed by Authors, complex relationships can be understood through network analysis, usually using bioinformatics to help elucidate mechanisms between drugs and their targets, from a pharmacokinetics and pharmacodynamics point of view. Specifically, transcriptome analysis results provide potential insights into tumor therapeutics in KYNA-treated AGS cells. In fact, it was found that Kynurenic acid induces the expression 349 of genes that induce apoptosis, and, above all, increases the expression of AP-1 factors.

However, it could be always to underline that these data derive from in vitro studies and lacks some fundamental pharmacokinetics and pharmacodynamics influences. So, could be premature to discuss about potential therapeutic role of Kynurenic acid in cancer.

Moreover, it could be important to discuss on pharmacokinetics of Kynurenic acid to understand the choice of the adopted concentrations.

Author Response

I am touched by your attentive and thoughtful review comments. It has been very helpful for my future research activities. Thank you very much.

*Please see the attachment file*

Reviewer 2 Report

In this work, Kim and co-authors performed a transcriptomic analysis on AGS cells in presence or absence of Kynurenic Acid (KYNA). Further, they investigated DEGs in KYNA-treated and untreated cells and they performed GO and protein-protein interactome by different bioinformatic tools. They identified  that KYNA treatment reduced AP-1 factors impacting on apoptosis. 

The work is in line and is similar to another work from the same research group recently pubblished (ref 24) that limits the novelty of the study. Moreover, the effect of KYNA treatment in cancer is controversal and still under debate (https://doi.org/10.1007/s00018-019-03332-w) and authors should have argued in details this specific issue. 

Moreover this reviewer is a bit confused about RNA-seq analysis including the specific data and related validation of AP-1 genes. Authors described in lines 255-258 "Western blot analysis of AGS cells during KYNA treatment confirmed the expression of c-Fos and c-Jun, which are factors AP-1 (Figure 5). It was confirmed that the expression of both proteins was decreased in a concentration-dependent manner when KYNA was treated." but the WB showed in fig. 5 did not confirm the abovementioned down-modulation. Transcriptomic data investigated gene transcripts and not PTMs. 

To run a transcriptomic experiment and related analysis is a good starting point for a study. Specifically, it is not clear here the number of analysed samples (2 per condition??), statistical analyses and the importance of all results coming out from sequencing. Moreover, the possibility to couple an additional (or more than one) gastric cancer cell line at least for validation experiments could have helped to enhance result importance and consistency.

Author Response

I am touched by your attentive and thoughtful review comments. It has been very helpful for my future research activities. Thank you very much.

Thank you for your attentive comments.  

  • In this work, Kim and co-authors performed a transcriptomic analysis on AGS cells in presence or absence of Kynurenic Acid (KYNA). Further, they investigated DEGs in KYNA-treated and untreated cells and they performed GO and protein-protein interactome by different bioinformatic tools. They identified that KYNA treatment reduced AP-1 factors impacting on apoptosis.
  • The work is in line and is similar to another work from the same research group recently pubblished (ref 24) that limits the novelty of the study. Moreover, the effect of KYNA treatment in cancer is controversal and still under debate (https://doi.org/10.1007/s00018-019-03332-w) and authors should have argued in details this specific issue.

Response : Thank you for your comment. Another study (ref 24) is a manuscript on the induction of apoptosis by treatment of KYNA with AGS, a gastric cancer cell line. However, this manuscript that studies the effects of kynurenic acid on the cell and the changing gene through the analysis of the transcript, which is the collective name for RNA expressed from DNA. Therefore, although the study on the mechanism of KYNA on gastric cancer cell lines is consistent, it is a completely different study from the study (ref 24) that confirmed the mechanism of protein expression and cancer cells. In addition, the fact that the biological activity of KYNA has not been clearly established shows a clear difference depending on the type of cancer and the cell line. If KYNA can have a negative effect on certain cancer cells, in this study, KYNA induces apoptosis in cancer cells. Therefore, further studies on other tumor cell lines are needed to confirm this. Additionally, after 2020, research papers that KYNA induces tumor cell death at appropriate concentrations have been continuously published. 1) ref - Apoptosis-mediated anticancer activity in prostate cancer cells of a chestnut honey (Castanea sativa L.) quinoline–pyrrolidine gamma-lactam alkaloid. 2021. Springer.

2) ref - A review of chromatographic methods for bioactive tryptophan metabolites, kynurenine, kynurenic acid, quinolinic acid, and others, in biological fluids. 2022. Weiley.

  • Moreover this reviewer is a bit confused about RNA-seq analysis including the specific data and related validation of AP-1 genes. Authors described in lines 255-258 "Western blot analysis of AGS cells during KYNA treatment confirmed the expression of c-Fos and c-Jun, which are factors AP-1 (Figure 5). It was confirmed that the expression of both proteins was decreased in a concentration-dependent manner when KYNA was treated." but the WB showed in fig. 5 did not confirm the abovementioned down-modulation. Transcriptomic data investigated gene transcripts and not PTMs.

Response : Thank you for your comment. In fig. 5 c0jun and c-fos act as housekeeping proteins as their total forms. Only when p-c-jun and p-c-fos decreased can we say that they decreased. In table 4, the genetic decrease of the AP-1 family was confirmed, and the expression of the protein was additionally confirmed. These methods suggest that changes in the transcript are actually effective by inducing changes in proteins.

  • To run a transcriptomic experiment and related analysis is a good starting point for a study. Specifically, it is not clear here the number of analysed samples (2 per condition??), statistical analyses and the importance of all results coming out from sequencing. Moreover, the possibility to couple an additional (or more than one) gastric cancer cell line at least for validation experiments could have helped to enhance result importance and consistency.

Response : Thank you for your comment. Thank you for noticing that running transcriptome experiments and related assays is a good starting point for research. Also, we know that your suggestion of adding one cell line will help increase the significance and consistency of the results. However, it takes a lot of time and money to conduct this additional study. In response to your comments, in the next study, we will proceed with the study by combining two or more cell lines. Thank you very much.

Thanks you for your detail comment.

*Please see the attachment file for the figure*

Round 2

Reviewer 2 Report

Comment to response 2) A phosphorylation is a PTM and correlate with TF activity. Transcriptomic analysis identified mRNA fluctuations according experimental conditions. Normally, in case of transcript downregulation the total protein decreases and this is not the case although author manuscript sentence and the specific comments here. Moreover, the profile of the total protein is strange. In general (and also in the datasheet of selected antibodies) in the total protein a double band (phosphorylated form) is visible. This reviewer remained not convinced about the specific WB analysis. The full membrane acquisition, gel information and molecular weight indication could help with data interpretation.

Comment to response 3) Due the heterogenicity of intrinsic features of cancer cell line (i.e. mutation status, stage of tumor, etc…) more than one cancer cell line are in general required to validate specific findings.